# Wavepacket insights into the photoprotection mechanism of the UV filter methyl anthranilate

Natércia d.N. Rodrigues[1], Neil C. Cole-Filipiak [1,3], Karl N. Blodgett [2], Chamara Abeysekera[2,4], Timothy S. Zwier[2] & Vasilios G. Stavros[1]

Meradimate is a broad-spectrum ultraviolet absorber used as a chemical filter in commercial sunscreens. Herein, we explore the ultrafast photodynamics occurring in methyl anthranilate (precursor to Meradimate) immediately after photoexcitation with ultraviolet radiation to understand the mechanisms underpinning Meradimate photoprotection. Using time-resolved photoelectron spectroscopy, signal from the first singlet excited state of methyl anthranilate shows an oscillatory behavior, i.e., quantum beats. Our studies reveal a dependence of the observed beating frequencies on photoexcitation wavelength and photoelectron kinetic energy, unveiling the different Franck-Condon overlaps between the vibrational levels of the ground electronic, first electronic excited, and ground cationic states of methyl anthranilate. By evaluating the behavior of these beats with increasing photon energy, we find evidence for intramolecular vibrational energy redistribution on the first electronic excited state. Such energy redistribution hinders efficient relaxation of the electronic excited state, making methyl anthranilate a poor choice for an efficient, efficacious sunscreen chemical filter.

[1] Department of Chemistry, University of Warwick, Coventry CV4 7AL, UK. [2] Department of Chemistry, Purdue University, West Lafayette, IN 47907, USA. [3] Present address: Combustion Research Facility, Sandia National Laboratories, Mail Stop 9055, Livermore, CA 94551-0969, USA. [4] Present address: Intel Corporation, Hillsboro, OR 97124, USA. Correspondence and requests for materials should be addressed to N.C.C-F. (email: ncolefi@sandia.gov) or to V.G.S. (email: v.stavros@warwick.ac.uk)

Ultraviolet (UV) radiation is vital for sustaining life, facilitating photosynthesis in plants[1] and production of vitamin D in humans[2], for example. Nevertheless, UV radiation can also have damaging effects[3,4], such as highly mutagenic DNA photolesions which may lead to skin cancer[5]. Human skin is naturally protected against radiative stress by melanin—dark pigments which absorb UV radiation before it reaches the DNA in skin cells[6]. However, the photoprotection afforded by melanin is often insufficient and sunburn may occur when skin is exposed to UV radiation[7]. Artificial skin protection against UV radiation is currently provided by commercially available sunscreen lotions, resulting in a recent interest in the UV initiated photodynamics of sunscreen molecules (see references[8–10] and references therein). Following photoexcitation, an ideal sunscreen molecule will rapidly (on a sub-ns timescale) undergo internal conversion to the ground electronic state, where the excess energy (now localized in the molecular vibrations) may be dissipated as heat.

In a previous study, we have reported the photodynamics of methyl 2-aminobenzoate (methyl anthranilate, MA), the precursor to the US FDA approved sunscreen component Meradimate (menthyl anthranilate, MenA)[11]. In that work, irradiating MA in the 330–300 nm range was found to access the first singlet excited state (S$_1$) resulting in hydrogen dislocation along the intramolecular N–H···O hydrogen bond. Evidence for one conical intersection (CI) accessible from the S$_1$ state of MA was found, though the CI was located beyond a barrier composed of a H-atom migration and subsequent internal rotation around the tautomer C–C double bond, as shown schematically in Fig. 1. Thus, the barrier to the CI effectively "trapped" population (hindered the non-radiative decay) on the S$_1$ state for ≫ns[11].

While photoexcitation above the barrier increased the excited state decay rate, the overall decay lifetime remained longer than the 1.2 ns experimental window[11]. At pump photon energies greater than the excited state barrier, substitution of the amine hydrogens with deuterium revealed a measurable kinetic isotope effect, suggesting that migration of the N–H···O hydrogen was the "rate determining step" in the overall decay dynamics.

Though no direct evidence for any energy redistribution on the excited state was found, excited state intramolecular vibrational energy redistribution (IVR) presumably hinders efficient coupling to the overall reaction coordinate shown in Fig. 1. To better understand the mechanisms behind the trapping of S$_1$ excited state population, we have studied the energy redistribution processes occurring in the first few picoseconds after photoexcitation of MA using time-resolved photoelectron spectroscopy (TR-PES)[12,13]. Through electron kinetic energy (eKE) and angular distributions, TR-PES carries information on both electronic configuration and vibrational dynamics and is thus a powerful tool for mapping energy flow in polyatomic molecules[13,14].

A typical TR-PES experiment will follow a pump-probe approach. First, the molecule of interest is photoexcited by a spectrally broad pump laser pulse ($\lambda_{pu}$), preparing a coherent superposition of $n$ eigenstates usually termed a wavepacket. The wavepacket then evolves freely in time according to:[12,15]

$$| \Psi(\Delta t) \rangle = \sum_n \tilde{a}_n |\psi_n\rangle e^{-i2\pi c E_n \Delta t} \qquad (1)$$

with $\tilde{a}_n$ accounting for both amplitudes (i.e. populations) and initial phases of the molecular eigenstates $|\psi_n\rangle$ with energy $E_n$ (presented here in wavenumber units). After some time delay ($\Delta t$), the probe laser pulse ($\lambda_{pr}$) projects the wavepacket onto a final state—in the case of TR-PES, a cationic state—generating a measurable photoelectron signal. Interference between the transitions from the different wavepacket eigenstates, $n$ and $m$, onto the same final state causes an oscillating modulation in the time-dependent photoelectron signal at frequencies $\omega = (E_n - E_m)/\hbar$, a phenomenon commonly referred to as quantum beats. Molecular quantum beat spectroscopy thus provides detailed information regarding intramolecular state couplings, as well as the dynamics of energy redistribution in polyatomic molecules[16–20].

In the present work, the observation of quantum beats in the TR-PES of MA is explored to probe the energy redistribution mechanisms immediately following the absorption of UV radiation. We aim to further elucidate the mechanism by which excited state population is trapped on the S$_1$ state. In addition, we briefly present quantum beat results from MenA in order to evaluate the effects of the additional menthyl group. Ultimately, excited state IVR appears to hinder efficient sampling of the S$_1$/S$_0$ CI, thus decreasing the non-radiative S$_1 \rightarrow$ S$_0$ relaxation rate of MA and reducing its efficiency and desirability as a photoprotection agent.

## Results

**Laser induced fluorescence and calculations.** The LIF spectrum, shown in Fig. 2a, is in good agreement with a previous measurement[21]. The vibrational progressions observed in the LIF spectra of both MA and its carboxylic acid variant anthranilic acid (AA) are also similar;[22] in MA, at least two prominent vibrational progressions are observed at 179 cm$^{-1}$ and 421 cm$^{-1}$. As shown in Fig. 2b, the 179 cm$^{-1}$ vibrational mode (henceforth referred to as $\nu_{179}$) corresponds to an in-plane bend of the ester group, while the 421 cm$^{-1}$ vibrational mode ($\nu_{421}$) corresponds to a bending motion of the carbonyl and amino groups with some distortion of the phenyl ring; similar motions were assigned to the LIF active modes in AA[22]. As shown in Supplementary Fig. 1, both modes show displacement upon photoexcitation, explaining the observed LIF progression. Several satellite features are present in the spectrum near each fundamental mode; an enlarged spectrum is shown in Supplementary Fig. 2. These features, not reproduced by electronic structure calculations (see Supplementary Table 1), are likely combination bands and/or Fermi resonances (see, for example, reference[23]).

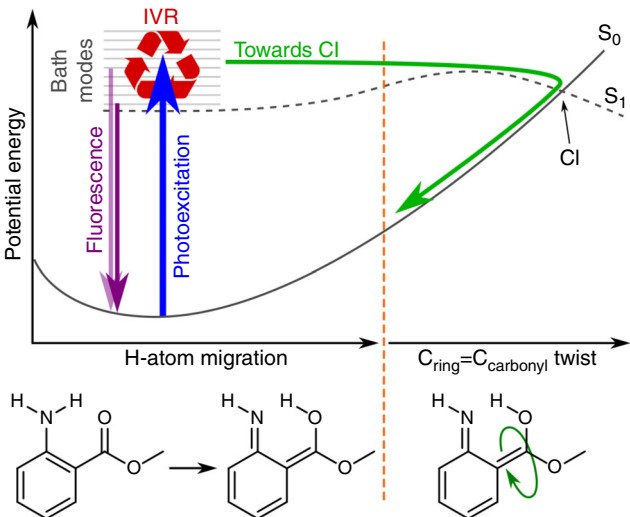

**Fig. 1** Photodynamics of methyl anthranilate (MA). Schematic potential energy cuts for the first electronic singlet excited state (S$_1$) and electronic ground state (S$_0$) of MA are shown, highlighting the dynamic pathways available to photoexcited MA (adapted from reference[11]). Excited state H-atom migration has a relatively small barrier to the tautomeric form followed by a much larger barrier along the out-of-plane C=C twist coordinate. The tautomer of MA is not a local minimum on the S$_1$ surface

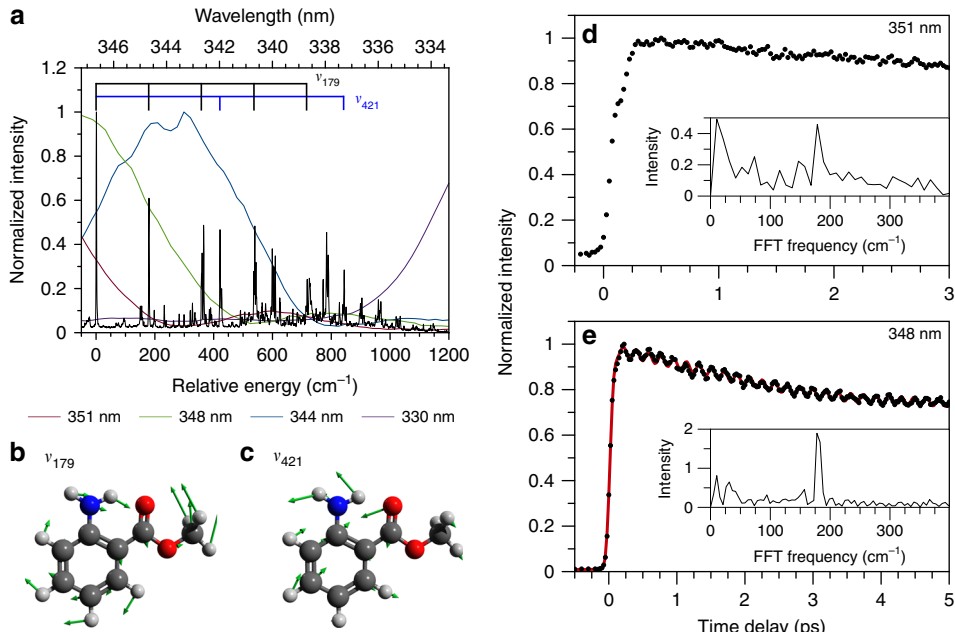

**Fig. 2** Spectra and vibrational modes of methyl anthranilate (MA) along with pump laser spectral profiles. **a** Laser Induced Fluorescence (LIF) spectrum of MA (black line) relative to the first singlet electronic excited state ($S_1$) absorption onset at 28 851 cm$^{-1}$. Overlaid are the spectral profiles of the pump laser pulses used in the time-resolved experiments. Shown below are vector displacement diagrams for (**b**) $\nu_{179}$ and (**c**) $\nu_{421}$ on the $S_1$ state. Example time-resolved photoelectron spectroscopy (TR-PES) transients at (**d**) 351 nm and (**e**) 348 nm pump wavelengths. Data are shown as black circles with the red line showing a fit. Insets show the corresponding fast Fourier transform (FFT) spectra

Having established the vibrational modes that are active in the close-to-origin photoexcitation of MA, it is important to specify which levels of these modes are accessed by the large bandwidth (~500 cm$^{-1}$ full width at half maximum) laser pulses used in our time-resolved experiments. As shown in Fig. 2a, at $\lambda_{pu} = 351$ nm the laser pulse accesses mainly the $S_1$ ($\nu = 0$) level of MA while $\lambda_{pu} = 348$ nm also accesses the $\nu = 1$ and 2 levels of $\nu_{179}$. At $\lambda_{pu} = 344$ nm, a much more complex combination of vibrational levels spanning over both progressions is accessed. Finally, at $\lambda_{pu} = 330$ nm, a highly vibrationally excited mixture of states is accessed (~1400 cm$^{-1}$ above the origin and near 300 vibrational states/cm$^{-1}$; see Supplementary Fig. 3).

**Time-resolved photoelectron spectroscopy.** The TR-PES transients of MA are obtained by integrating an eKE region of interest (typically between ~150–2500 cm$^{-1}$) as a function of time delay; examples of the resulting 1-D intensity vs. time delay transients are shown in Figs. 2 and 3. The 1-D transient at $\lambda_{pu} = 348$ nm, for example, reveals signal oscillations and provides clear evidence for quantum beats in MA. The transients were analyzed with a fast Fourier transform (FFT) algorithm, as described in reference[20], to reveal the frequencies of the quantum beats (shown as insets in Fig. 2d, e). Transients were also fit using kinetic models that include a sum of sine waves, the frequencies of which were compared to the FFT results to ensure agreement; an example of such fit at $\lambda_{pu} = 348$ nm is shown in Fig. 2e and fit parameters are given in Supplementary Table 2. Similar analyses were performed on the full, 2-D TR-PES transients as a function of eKE for all time delays, yielding 2-D FFT vs. eKE spectra and highlighting the eKE at which a beat frequency appears— henceforth referred to as FFT(eKE). The evolution of the beat frequencies as a function of time delay was studied by producing FFT spectra for a given eKE range as a function of time delay, yielding FFT($\Delta t$) spectra similar to reference[24]. Further discussion of all data processing and fitting procedures may be found in

the Supplementary Methods. Example transients at different fitting stages are shown in Supplementary Fig. 9; a schematic representation of the production of FFT(eKE) and FFT($\Delta t$) spectra is also given in Supplementary Fig. 10.

**351 and 348 nm photoexcitation.** As expected from the spectra in Fig. 2, the observed quantum beat frequencies are pump wavelength dependent. At the lowest energy photoexcitation, $\lambda_{pu} = 351$ nm, no oscillations are observed in the transient (see Fig. 2d), suggesting that this pump energy is only populating the ground vibrational state of $S_1$. The near-negligible feature at ~180 cm$^{-1}$ in the FFT spectrum is likely due to the edge of the 351 nm pulse exciting the $\nu = 1$ level of $\nu_{179}$, as shown in Fig. 2a. At $\lambda_{pu} = 348$ nm, a prominent beat frequency of ~180 cm$^{-1}$ is extracted from both the FFT spectrum and sine wave fit to the TR-PES transient, consistent with the excitation of the $\nu = 0$, 1, and 2 levels of $\nu_{179}$ shown in Fig. 1e. An additional frequency of *ca.* 5 cm$^{-1}$ (see Supplementary Table 2) is also present due to the short wavelength edge of the pump laser spectrum interacting with higher lying vibrational states. The oscillations at 348 nm show little evidence of dampening or decay by 5 ps, supporting the expectation that population in the low-lying levels of $\nu_{179}$ persists for much longer than the duration of these experiments;[25] FFT (eKE) and FFT($\Delta t$) spectra are shown in Supplementary Fig. 4.

**330 nm photoexcitation.** At $\lambda_{pu} = 330$ nm, the longest wavelength used in our previous investigation[11], the FFT(eKE) spectrum in Fig. 3 shows features around 60, 115, and 180 cm$^{-1}$. As before, the 180 cm$^{-1}$ beat frequency may be assigned to interference between levels of $\nu_{179}$. While the present LIF spectrum does not extend to 330 nm, 60 and 115 cm$^{-1}$ energy spacings arise from multiple combinations of vibrationally excited $\nu_{179}$ and $\nu_{421}$, thus suggesting that the 60 and 115 cm$^{-1}$ beats arise from combination bands involving these two modes. The observed beats are dependent on eKE: the 180 cm$^{-1}$ beat appears closer to

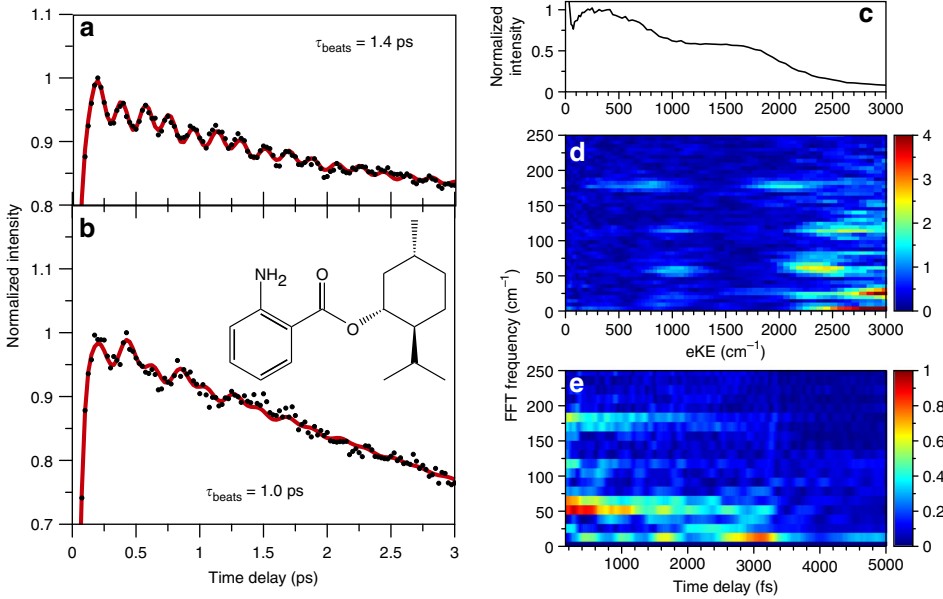

**Fig. 3** Example transients, spectra and analysis for MA and MenA at $\lambda_{pu}$ = 330 nm. Example time-resolved phoroelectron spectroscopy (TR-PES) transients at a pump wavelength ($\lambda_{pu}$) of 330 nm for (**a**) methyl anthranilate (MA) and (**b**) menthyl anthranilate (MenA). Data are shown as black circles with fits shown as a red line. Inset: structure of MenA. **c** The photoelectron spectrum of MA at $\lambda_{pu}$ = 330 nm. Results of the fast Fourier transform (FFT) analysis including (**d**) FFT spectra as a function of electron kinetic energy (eKE) and (**e**) FFT($\Delta t$) for eKE ≈ 300–2900 cm$^{-1}$

each photoelectron intensity maximum while the high eKE edges of each photoelectron feature reveal evidence of interference with $v_{421}$. The 180 cm$^{-1}$ beat frequency also shows a ~$\pi$/2 phase shift relative to the other beat (see Supplementary Table 2). An additional beat frequency of ≤ 25 cm$^{-1}$ is also observed at $\lambda_{pu}$ = 30 nm, however this beat is likely from interferences between vibrational modes not captured in the LIF spectrum. Indeed, excited state frequency calculations reveal five modes between 1400 and 1500 cm$^{-1}$ (near the center of the 330 nm pump spectrum), generally described as in-plane ring distortions.

The quantum beats observed at $\lambda_{pu}$ = 330 nm have a significantly shorter decay lifetime than at $\lambda_{pu}$ = 348 nm; a timescale of 2–3 ps may be estimated from the FFT($\Delta t$) results in Fig. 3e. As shown in Supplementary Fig. 5, measurements extending to $\Delta t$ = 12.5 ps show no evidence for quantum beat revival. The shorter decay times at $\lambda_{pu}$ = 330 nm allow for the 1-D transient to be fit with a kinetic model (see Fig. 2a) yielding an overall beat decay lifetime of 1.4 ± 0.2 ps; the reported error is the standard error from the fit. Thus, FFT($\Delta t$) intensity changes appear to be representative of beat dampening—this assumption will be made throughout the ensuing discussion.

A one-dimensional transient at $\lambda_{pu}$ = 330 nm was also obtained for MenA; the results are also shown in Fig. 3b. Upon replacing the methyl group with the larger menthyl group both the beat pattern and dampening time in MenA are altered with the fit returning a slightly shorter dampening lifetime of 1.0 ± 0.3 ps.

**344 nm photoexcitation**. The beat behavior of MA at $\lambda_{pu}$ = 344 nm was also investigated; an example isolated transient (i.e. after normalizing to the overall decay, as discussed in the Supplementary Methods) is shown in Fig. 4a. Despite the different vibrational modes excited by this pump wavelength (see Fig. 2a), the FFT(eKE) spectra at this wavelength reveal similar quantum beat frequencies to both the 348 and 330 nm results; see Supplementary Fig. 6. At $\lambda_{pu}$ = 344 nm, the 60 and 115 cm$^{-1}$ beat frequencies may be firmly assigned to interference between the $v$

= 1 level of $v_{421}$ with the $v$ = 2 and 3 levels of $v_{179}$ (a combination not directly excited by the 330 nm pump laser). As can be seen in Supplementary Fig. 6, the beat previously observed at 60 cm$^{-1}$ now comprises two distinct features at 62 and 56 cm$^{-1}$ due to an additional interference term between the $v$ = 1 level of $v_{421}$ and the prominent LIF peak at 366 cm$^{-1}$ (see Supplementary Fig. 2). As was the case at $\lambda_{pu}$ = 330 nm, the beats at $\lambda_{pu}$ = 44 nm still show a clear eKE dependence with vibrational excitation retained in the $v_{179}$ mode upon ionization (shown in Supplementary Fig. 6) and a relative phase of approximately $\pi$/2 for the 180 cm$^{-1}$ beat. Indeed, as shown in the two-dimensional transient in Supplementary Fig. 7, each beat shows a clear temporal offset with decreasing eKE.

Two $\lambda_{pu}$ = 344 nm transients were acquired over two successive and overlapping time windows; the resulting FFT($\Delta t$) spectra are shown in Fig. 4b, c. Both the transient and FFT($\Delta t$) spectra reveal a much richer temporal behavior than at the other wavelengths studied. First, the quantum beats near 60 and 115 cm$^{-1}$ appear to dampen within the first two picoseconds. The 60 cm$^{-1}$ beats, however, show a revival at ~5 ps. This revival is more apparent in Fig. 4c, where the 60 cm$^{-1}$ FFT intensity is larger relative to the other intensities (due to the intensity scaling of the FFT analysis). The 180 and 115 cm$^{-1}$ beats appear to show revivals at 7 and 9.5 ps, respectively.

**Wavepacket calculations**. To aid the interpretation of these transients, the time-autocorrelation function of Eq. (1) was calculated[15] using the dominant transitions observed in the LIF spectrum; see Supplementary Methods and Supplementary Table 4 for further details. Essentially, the time-autocorrelation function describes the temporal behavior of the wavepacket as it moves out of and back into the Franck-Condon region. Plots of the survival probability of the time-autocorrelation function, i.e. $|\langle\Psi(\Delta t)|\Psi(0)\rangle|^2$, may serve as an initial comparison to experimental transients[26,27]; an example of the calculated "transient" is shown in Fig. 4d. The corresponding FFT($\Delta t$) plots from these wavepacket calculations are shown in Fig. 4e, f and

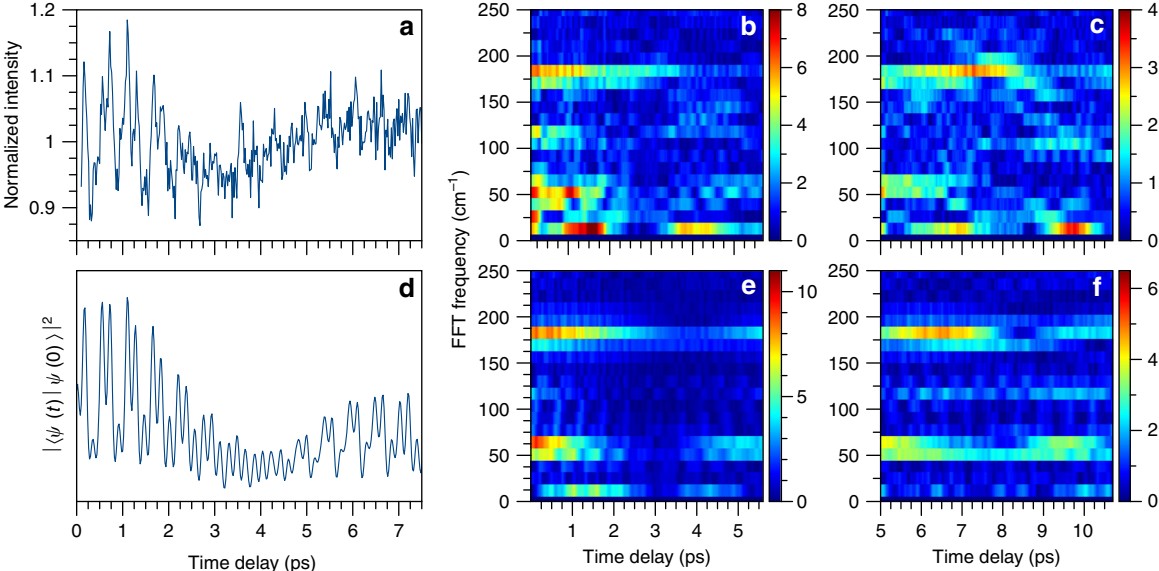

**Fig. 4** Results at $\lambda_{pu} = 344$ nm: example transients, FFT($\Delta t$) spectra, and time-autocorrelation function calculations. **a** Example isolated transient at $\lambda_{pu} = 344$ nm. The corresponding FFT($\Delta t$) for (**b**) $\Delta t = 0$–7.5 ps and (**c**) $\Delta t = 5$–12.5 ps are also shown. The weaker FFT intensities around 5 ps in **b** are clearer in **c** due to the relative intensity scaling of the FFT analysis. **d** Time-autocorrelation function calculated from Eq. (S7) and the LIF spectra at $\lambda_{pu} = 344$ nm (see Supplementary Methods for details). The corresponding FFT($\Delta t$) from the time-autocorrelation function over the same temporal windows as **b** and **c** are presented in **e** and **f**, respectively

Supplementary Figure 12. The calculated behavior shows remarkably good agreement with the experimental beats, reproducing the overall time behavior of the isolated 1-D transient and FFT spectra, excepting a spurious revival of the 60 cm$^{-1}$ beat at ~9.5 ps. To achieve good agreement with the experimental transient at 344 nm, a vibrational mode dependent phase of $\pi/2$ was added to the LIF peaks associated with $\nu_{179}$ (see Supplementary Table 3).

## Discussion

The agreement between the dominant LIF active modes and the extracted beat frequencies is compelling evidence for the assignment of the observed beats to the $\nu_{179}$ and $\nu_{421}$ modes with contributions from nearby bands. Interestingly, this agreement also suggests that the observed beats are due solely to "bright" ($S_1 \leftarrow S_0$ active) states. The beat frequency dependence upon eKE reveals that vibrational excitation of the $S_1$ state is preserved in $\nu_{179}$ upon ionization and confirms that the 180 cm$^{-1}$ beat frequency is largely due to interference between multiple excited levels of $\nu_{179}$. Beats involving the $\nu = 1$ level of $\nu_{421}$, however, only appear at the highest eKE as the $\nu = 0$ level of the $D_0^+$ state is the common final state onto which all vibrational modes may project. The absence of beat frequencies larger than ~250 cm$^{-1}$ is due to the temporal resolution of our experiment (see Supplementary Fig. 11 in the Supplementary Methods). Further discussion of the photoelectron structure and eKE dependence of the quantum beats may be found in Supplementary Discussion 1.

With vibrational assignments to the dominant quantum beats, we now focus on the pump wavelength dependent decay times of these beats. For large polyatomic molecules, the disappearance of quantum beats is due to depopulation of the initially prepared vibrational states via intramolecular vibrational energy redistribution (IVR)[28–30]. At $\lambda_{pu} = 348$ nm, where only the lowest vibrational levels of MA are populated, the 180 cm$^{-1}$ beat is a signature feature of a coherent superposition of the $\nu = 1$ and $\nu = 0$ levels of $\nu_{179}$. As there is no "bath" of modes at this excitation energy, see Supplementary Fig. 3, IVR does not occur and

population remains localized in these modes for the duration of the experiment. Conversely, at $\lambda_{pu} = 330$ nm a rapid dampening of the beats is observed as the much higher density of states facilitates rapid, "statistical" IVR (provided efficient coupling between states is maintained)[25].

In our previous time-resolved study, we found the excited state population of MA to be trapped in the $S_1$ state, incapable of overcoming a ~0.4 eV barrier to a $S_1/S_0$ CI via a twisting motion of the ester group[11]. Our present results at $\lambda_{pu} = 330$ nm suggest that the long excited state lifetime observed at shorter wavelengths (i.e. above the excited state barrier) is likely due to IVR following photoexcitation; low-lying "vibration-torsion" doorway states[19,25,30,31] could facilitate the coupling of the initially excited vibrational modes to a non-reactive bath of modes[32,33]. Indeed, even at relatively low vibrational state densities, molecules with methyl rotors can show complex vibrational state interactions[32–34]. Such excited state energy randomization would thus result in a "statistical-like," i.e. nanosecond, depopulation of the MA $S_1$ state (as was previously observed[11]) via fluorescence and/or the $S_1/S_0$ CI. As seen in Fig. 3b, the addition of the menthyl group (and the resulting increased density of states) in MenA appears to decrease the IVR lifetime, explaining the qualitatively slower decay of the $S_1$ state of this sunscreen agent observed in our previous work[11].

At the photon energies needed to overcome the excited state barrier in MA, IVR on the electronic excited state is unsurprising. Similarly at $\lambda_{pu} = 344$ nm, ~200 cm$^{-1}$ above the $S_1$ origin and at approximately 0.2 vibrational states/cm$^{-1}$ (see Supplementary Fig. 3), we do not expect to see any IVR-like behavior[25]. Indeed, the quantum beats at this wavelength show clear evidence for dephasing and revival with the time-autocorrelation calculation in qualitatively good agreement with the revival times of each frequency. The most significant disagreement, an extra revival of the 60 cm$^{-1}$ beat, is likely due to inaccuracies in the phase factor added to each vibrational level in the calculation. As discussed further in Supplementary Discussion 2, the inclusion of a phase factor into the wavepacket calculation is crucial to the

reproduction of the experimental transients and observed quantum beat phase (see Supplementary Table 2).

The origin of this phase factor and the experimentally observed quantum beat phase may be understood by considering the appearance of the $180\,cm^{-1}$ beat at lower eKE. While the energy shift is related to the projection of $\nu_{179}$ into the vibrationally excited cation (as discussed above), the diagonal beat recurrences shown in Supplementary Fig. 7 show that quantum beats appearing at lower eKE also appear to be shifted in time delay. Such a temporal offset may be described as a quantum beat (namely the $180\,cm^{-1}$ beat) with a different phase[15]. As this phase is tied to a change in eKE, the phase factors added to the time-autocorrelation function must serve to independently shift the time-autocorrelation "probe window" along each vibrational coordinate involved in a given beat. Furthermore, the inclusion of a phase factor into these wavepacket calculations that depends on specific vibrational modes implies that a majority of the modes contributing to the $180\,cm^{-1}$ beat are observed away from their respective classical turning points (i.e. the potential energy walls of each vibrational mode, see reference [35] for example) in the experiment. While the mixed nature of the $S_1$ vibrational states (as evidenced by the large number of satellite peaks near the main LIF progressions) makes firm assignment of each vibrational phase untenable at present, we may yet unveil some information about the $S_1$ state. A reasonable wavepacket reproduction of the experimental transient is obtained when the levels of $\nu_{179}$ are given a phase factor of $\pi/2$ relative to the other vibrational modes (as shown in Fig. 4d). This $\nu_{179}$ phase factor is different from applying a phase of $\pi/2$ to any $\sim180\,cm^{-1}$ beat, as several other combinations of LIF peaks also contribute to this beat (most significantly the LIF peak at $366\,cm^{-1}$ and the cluster of peaks near $600\,cm^{-1}$ shown in Supplementary Fig. 2). Indeed, as shown in Supplementary Fig. 8, wavepacket calculations with a phase factor of $\pi/2$ added to these modes result in substantially poorer agreement to the experimental 344 nm transient. It thus appears that the ionization window along the $\nu_{179}$ coordinate is shifted relative to the other normal modes.

Regardless of the nature of the observed quantum beat phase, the time-autocorrelation function is too simplistic to accurately model the MA beat behavior: this function does not contain any information about the $S_1$ potential energy landscape or an explicit projection onto the $D_0^+$ surface, factors which must play a significant role in the experimentally observed temporal behavior of these beats. A more accurate representation of photoexcitation and time-dependent photoionization of MA (such as through direct dynamics calculations) may reproduce the observed beat behavior[36,37] and would be of significant interest. Thus, there is a clear need for additional theoretical and experimental work (both time-resolved vibrational and frequency-resolved vibronic spectroscopies) to elucidate the nature of the observed wavepacket behavior and further our understanding of vibrational energy flow in MA.

In summary, the study reported herein has provided insight into the vibrational energy redistribution processes occurring in MA (a sunscreen precursor) immediately after photoexcitation with UV radiation. The LIF and quantum beats provide insight into the Franck-Condon activity of the $S_1 \leftarrow S_0$ transition, as well as the relative geometries of the $S_1$ and $D_0^+$ states. Our results have unveiled the ultrafast IVR processes in MA at $\lambda_{pu} = 330$ nm which allow for prompt redistribution of population on the $S_1$ state, further elucidating the trapping mechanism of excited state population of MA at shorter wavelengths. We also present evidence for a vibrational mode dependent phase not easily reproduced by a time-autocorrelation model, suggesting that further exploration is necessary to understand time-dependent ionization in the analysis of molecular quantum beats.

By understanding the fate of energy absorbed by an isolated sunscreen precursor we can reveal the key mechanisms that afford – or hinder – effective photoprotection to sunscreen active ingredients. In the case of MA, while the intramolecular hydrogen bond affords the molecule photostability (maintaining molecular integrity following photoexcitation), the same moiety also stabilizes the excited state and hinders rapid, non-radiative decay[11]. As H-atom migration, a key step towards the $S_1/S_0$ CI in MA[11], does not appear to be an active mode with UV-A photoexcitation (400–315 nm), the bath of modes available to MA and MenA instead hinders efficient, rapid excited state evolution towards the CI and thus impedes ideal photoprotection. As such, this work supports our previous conclusion that MA and MenA are a poor choice for an efficient sunscreen chemical absorber.

The results presented herein highlight an important consideration for rational sunscreen design: ensuring that the Franck-Condon active vibrational modes are involved in the rapid return to the ground electronic state, completing the absorption-recovery cycle of photoprotection prior to significant excited state energy randomization. Furthermore, it is important to extend these studies towards a "commercial" sunscreen scenario whereby the sunscreen molecule is made to interact with solvents and other sunscreen ingredients (such as reference [38]). Evaluating interactions in a complex environment, where energy transfer processes may dominate excited state relaxation, is essential to understanding the global mechanisms of photoprotection.

## Methods

**Laser induced fluorescence.** The laser induced fluorescence (LIF) apparatus is described in detail elsewhere[39]. A liquid sample of MA (Sigma-Aldrich, 99%) was placed in an unheated glass sample holder behind an 800 μm diameter nozzle (Parker General Valve series 9), entrained in ~3 bar He, and pulsed into a vacuum chamber at 20 Hz, creating a supersonic expansion that cools the molecules into their vibrational zero-point level prior to interrogation. Approximately 8 mm downstream from the nozzle, the frequency doubled output of a Nd:YAG-pumped dye laser (LambdaPhysik Scanmate) interacted with the expansion-cooled molecules in the focal point of two 4-inch diameter spherical mirrors, the bottom mirror containing a 1 cm hole through which the fluorescent photons were focused. These spherical mirrors steered the fluorescence photons through a 2-inch diameter plano-convex collimating lens, a long-pass filter to reduce scattered light, and onto a UV-sensitive photomultiplier tube (PMT).

The dye laser frequency was scanned through the 347–335 nm range that includes the $S_1 \leftarrow S_0$ origin transition ($0_0^0$) while collecting the total fluorescence. The fluorescence decay profile from the PMT was digitized by an oscilloscope (Tektronix, model 3052B), sent to a computer, and plotted as a function of excitation wavelength.

**Time-resolved photoelectron spectroscopy.** The TR-PES apparatus used in this work has been described previously[40–42] and consists of a commercially available femtosecond (fs) laser system, a molecular beam source and a photoelectron spectrometer[43]. A Ti:Sapphire oscillator (Spectra-Physics Tsunami) and regenerative amplifier (Spectra-Physics Spitfire XP) are used to produce ~40 fs laser pulses centered at 800 nm at ~3 mJ per pulse and 1 kHz. The fundamental 800 nm output is subsequently split into three 1 W beams. One of these beams is used to pump an optical parametric amplifier (Light Conversion TOPAS-C) to generate "pump" ($\lambda_{pu}$) pulses, with centers varied between 330–351 nm. A second 800 nm laser beam pumps a separate TOPAS-C which is used to generate "probe" ($\lambda_{pr}$) pulses, centered at 285 nm for MA and 296 nm for MenA. Probe wavelengths were chosen such that the total photon energy is slightly larger than the ionization energy of each molecule. The pump and probe pulses are temporally delayed with respect to each other ($\Delta t$) by a retroreflector mounted on a motorized delay stage, allowing a maximum temporal delay of 1.2 ns. Laser powers were set to ensure single photon initiated photochemistry.

The two laser beams intersect a molecular beam which is produced by seeding MA (Alfa Aesar, 99%) heated to 55 °C or MenA (Aldrich, 98%) heated to 90 °C into helium (~3 bar). This gas mixture is then expanded into vacuum using an Even-Lavie pulsed solenoid valve[23,24]. At $\Delta t > 0$, $\lambda_{pu}$ excites molecules entrained in the beam and $\lambda_{pr}$ ionizes the excited species. The resulting photoelectrons are focused onto a position-sensitive microchannel plate (MCP) detector coupled to a phosphor screen such that electrons with the same initial velocity are mapped onto the same point on the detector[43]. The original photoelectron distribution is reconstructed from the measured two-dimensional (2-D) projection using a polar onion peeling algorithm[44] from which the desired one-dimensional (1-D)

photoelectron spectrum was derived. The spectrometer is calibrated using the well-known photoelectron spectrum of photoionized Xe[45].

**Computational methods**. All calculations were performed using the GAMESS computational package[46,47]. The geometry of MA was optimized on both the $S_0$, $S_1$, and $D_0^+$ surfaces at the (TD-)CAM-B3LYP[48]/cc-pVDZ[49] level of theory[50–52]. Harmonic frequencies were calculated at the $S_0$ equilibrium ($S_{0,eq}$), $S_1$ vertical Franck-Condon ($S_{1,vFC}$), $S_1$ equilibrium ($S_{1,eq}$), $D_0^+$ vertical Franck-Condon ($D_{0,vFC}^+$), and $D_0^+$ equilibrium ($D_{0,eq}^+$) geometries. The vibrational density of states was calculated from the $S_{1,eq}$ frequencies using the direct counting method of Beyer and Swinehart[53].

The Franck-Condon active vibrational modes were identified from the LIF results. Potential energy cuts (PECs) along these normal modes were generated by a Cartesian coordinate offset from a starting geometry (e.g., $S_{0,eq}$) using the wxMacMolPlt program[54]. Single-point energies were then calculated at each set of offset coordinates and fit with a harmonic potential.

**Wavepacket calculations**. Detailed descriptions of the wavepacket calculations may be found in the Supplementary Methods. Briefly, the absolute square of the time-autocorrelation of Eq. (1), i.e. $|\langle\Psi(\Delta t)|\Psi(0)\rangle|^2$, was numerically evaluated by a MATLAB program. The results were then temporally smoothed to account for the pump and probe laser temporal widths, yielding one-dimensional calculated "transients." The wavepacket calculation code is freely available in the Zenodo data repository, as described below. The one-dimensional, calculated "transients" were subjected to the same FFT($\Delta t$) analysis as the isolated experimental transients as described in the Supplementary Methods.

## Data availability

The TR-PES, LIF, computational data, and wavepacket propagation program underpinning the present research are freely available in the Zenodo data repository with the identifier doi:10.5281/zenodo.1472500; descriptions of the data formats and further details are given in Supplementary Note 1.

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

## Acknowledgements
The authors would like to thank Dr. M. Staniforth for helpful discussions and sample code for FFT analysis. N.d.N.R. thanks the EPSRC for doctoral funding. N.C.C.F. thanks the Leverhulme Trust for postdoctoral funding. V.G.S. thanks the EPSRC for equipment grants (EP/J007153 and EP/N010825) and the Royal Society and Leverhulme Trust for a Royal Society Leverhulme Trust Senior Research Fellowship. K.N.B., C.A., and T.S.Z. gratefully acknowledge support from the Department of Energy Basic Energy Sciences, grant DE-FG02-96ER14656.

## Author contributions
N.d.N.R. and N.C.C.F. ran the time-resolved experiments, performed data analysis, and prepared the manuscript. N.C.C.F. ran the theoretical calculations. K.N.B., C.A., and T.S. Z. performed the frequency-resolved LIF measurements. V.G.S. conceived and designed the experiment, assisted in data analysis and interpretation, and write-up of the manuscript.

## Additional information

**Competing interests:** The authors declare no competing interests.

