## [Peer Review File · Nature Communications]

Reviewers' comments:

Reviewer #1 (Remarks to the Author):

Stavros et al present a time-resolved study of an isolated methyl anthranilate (MA), which is a precursor of component for sunscreen agents and thus a molecule with a photochemistry that is of wider interest. The group uses photoelectron spectroscopy as a probe method, which permits to gain the maximum insight into the dynamics. The authors observe quantum beats in the S₁ state and analyse the redistribution of vibrational energy. The experiments are carried out expertly, the data analysis is described in great detail in the S1 and the conclusions are convincing. Therefore I recommend to publish the manuscript as is.

I only have a few remarks:

1. I tried to look up the files in the Zenodo repository, but couldn't find them. Are they only available after publication, is there a mistake in the address or am I too stupid to find it? Please check!
2. The role of intramolecular hydrogen bonding for the dynamics should be discussed briefly.
3. In the conclusion the authors describe the key mechanisms that "afford or hinder" photoprotection. The authors should point out more clearly which photochemical properties afford and which properties hinder photoprotection. They should also conclude whether MA is a good photoprotection or not, based on their results.

Reviewer #2 (Remarks to the Author):

In this manuscript, Stavros and co-workers employ time-resolved photoelectron spectroscopy to provide important new insight on the photophysical dynamics of molecules that function as sunscreen agents, in this case, methyl anthranilate (MA). Their use of femtosecond UV pulses for photoexcitation leads to the generation of multimode vibrational wave packets, whose vibrational frequencies are in good agreement with those obtained from laser-induced fluorescence (LIF) spectroscopy. By tuning the UV excitation wavelength, and hence controlling the amount of excess energy deposited in the molecule, the authors observe varying vibrational dephasing times, which in turn report on the relative efficiency of intramolecular vibrational redistribution (IVR) for different excitation wavelengths. The authors find that IVR occurs on the 1-ps timescale with highest energy excitation (330 nm). The ultrafast redistribution of excess vibrational energy, which prevents access to the S₀-S₁ conical intersection, explains the long excited-state lifetime of MA. In addition, at intermediate levels of excitation (344 nm), the authors observe a multitude of vibrational

frequencies, some exhibiting picosecond revivals, which they skillfully analyze with the aid of the LIF spectrum and wave packet calculations. Given the important new insight that this work provides on the photophysical behavior of molecular sunscreens, and the high level of finesse with which the experiments (highly challenging!) are executed and the data are analyzed, I strongly recommend the publication of this work in Nature Communications. There are some optional minor comments that the authors might want to address should they be asked to resubmit a revised version of their manuscript:

1. To enhance the general readability of the manuscript, and to better inform the reader regarding the significance of their findings, the authors might wish to consider including a schematic illustration of the S0 and S1 potential energy surfaces in the introduction section of the manuscript, along with the various competing processes – IVR vs. internal conversion via conical intersection – occurring on S1.
2. In Figure 2d, the 180-cm⁻¹ oscillation component is peaked at eKE values of ~800 cm⁻¹ and ~2100 cm⁻¹, whereas the lower frequency components are peaked at higher eKE values. Could the authors please provide an explanation for the different behaviors?
3. In their time-domain analysis, the authors use a common damping time for all the vibrational frequencies. It seems counterintuitive to me that the dephasing time is independent of vibrational frequency. Have the authors considered using different dephasing times for different frequencies? Alternatively, can the authors provide a brief explanation for why dephasing times should be independent of vibrational mode?
4. In the Supplementary Information, it would be useful for the authors to list the frequencies extracted from their fits to eq. S3 (or S4), along with the amplitudes and phases.
5. For the non-experts, the authors may wish to briefly describe the physical significance of the time-autocorrelation function. On a related note, Figure 3d shows the plot of the computed autocorrelation function. However, shouldn't it be the modulus-squared of the autocorrelation function – the survival probability – that should be compared to the experimental time-dependent photoelectron signal instead?

Response to Reviewer Comments:

We have reproduced the reviewer comments below in blue. Our response to each reviewer comment then follows. Any changes to the manuscript have then been described and reproduced with changes highlight in yellow.

Response to Reviewer #1:

1. I tried to look up the files in the Zenodo repository, but couldn't find them. Are they only available after publication, is there a mistake in the address or am I too stupid to find it? Please check!

Author Response and Action: We would like to apologize for any confusion the reviewer may have experienced while reading the manuscript. The Zenodo repository allows authors to reserve a DOI without publishing the research data. We elected to withhold publication of these data until the manuscript had been accepted and assigned a DOI (which we will in turn add to the Zenodo archive). At the reviewer or editor's request, we can make these data available to the public sooner.

2. The role of intramolecular hydrogen bonding for the dynamics should be discussed briefly.

Author Response and Action: We have enhanced our discussion of the overall MA decay mechanisms in the Introduction. Paragraphs 2 and 3 of this section now read:

...In that work, irradiating MA in the 330–300 nm range was found to access the first singlet excited state (S_1) resulting in hydrogen dislocation along the intramolecular N–H···O hydrogen bond. Evidence for one conical intersection (CI) accessible from the S_1 state of MA was found, though the CI was located beyond a barrier composed of a H-atom migration and subsequent internal rotation around the tautomer C–C double bond, as shown schematically in Figure 1. Thus, the barrier to the CI effectively “trapped” population (hindered the non-radiative decay) on the S_1 state for \gg ns.¹¹

While photoexcitation above the barrier increased the excited state decay rate, the overall decay lifetime remained longer than the 1.2 ns experimental window.¹¹ At photon energies greater than the excited state barrier, substitution of the amine hydrogens with deuterium revealed a measurable kinetic isotope effect, suggesting that migration of the N–H···O hydrogen was the “rate determining step” in the overall decay dynamics. Though no direct evidence for any energy redistribution on the excited state was found, excited state intramolecular vibrational energy redistribution (IVR) presumably hinders efficient coupling to the overall reaction coordinate shown in Figure 1. To better understand the mechanisms behind the trapping...

3. In the conclusion the authors describe the key mechanisms that “afford or hinder” photoprotection. The authors should point out more clearly which photochemical properties afford and which properties hinder

photoprotection. They should also conclude whether MA is a good photoprotection or not, based on their results.

Author Response and Action: We have added some text to the Introduction and Conclusion sections highlighting some of the photochemical properties and processes of MA that afford and hinder photoprotection. We have also added Figure 1 to the introduction to more explicitly describe the overall photochemistry of MA. The text of the introduction now reads:

... further elucidate the mechanism by which excited state population is trapped on the S_1 state. In addition, we briefly present quantum beat results from MenA in order to evaluate the effects of the additional menthyl group. Ultimately, excited state IVR appears to hinder efficient sampling of the S_1/S_0 CI, thus decreasing the non-radiative $S_1 \rightarrow S_0$ relaxation rate of MA and reducing its efficiency and desirability as a photoprotection agent.

The text of the conclusion now reads:

...By understanding the fate of energy absorbed by an isolated sunscreen precursor we can reveal the key mechanisms that afford – or hinder – effective photoprotection to sunscreen active ingredients. In the case of MA, while the intramolecular hydrogen bond affords the molecule photostability (maintaining molecular integrity following photoexcitation), the same moiety also stabilizes the excited state and hinders rapid, non-radiative decay.¹¹ As H-atom migration, a key step towards the S_1/S_0 CI in MA,¹¹ does not appear to be an active mode with UV-A photoexcitation (400–315 nm), the bath of modes available to MA and MenA instead hinders efficient, rapid excited state evolution towards the CI and thus impedes ideal photoprotection. As such, this work supports our previous conclusion that MA and MenA are a poor choice for an efficient sunscreen chemical absorber.

Response to Reviewer #2:

1. To enhance the general readability of the manuscript, and to better inform the reader regarding the significance of their findings, the authors might wish to consider including a schematic illustration of the S_0 and S_1 potential energy surfaces in the introduction section of the manuscript, along with the various competing processes – IVR vs. internal conversion via conical intersection – occurring on S_1 .

Author Response and Action: The reviewer raises a key weakness here regarding readability of the manuscript; their suggestion is appropriate and well taken. Based on our previous work on MA, we have added Figure 1 to the main paper. Based on a previous publication from our group, this figure highlights the competing processes in MA as well as the relevant reaction coordinates for non-radiative decay of the S_1 state.

2. In Figure 2d, the 180-cm⁻¹ oscillation component is peaked at eKE values of ~800 cm⁻¹ and ~2100 cm⁻¹, whereas the lower frequency components are peaked at higher eKE values. Could the authors please provide an explanation for the different behaviors?

Author Response and Action: Briefly, the vibrational levels of the 180 cm⁻¹ vibrational mode show strong Franck-Condon overlap with the excited vibrational levels of the corresponding mode in the cation. Ionization of the wavepacket will thus result in a 180 cm⁻¹ beat frequency extending to lower eKE (and thus higher internal energy of the cation). More importantly, though, the other beat frequencies (*i.e.* not appearing at 180 cm⁻¹) appear at the highest eKE as the $v = 0$ level of these modes in the cation appear to be the only state onto which vibrationally excited levels of both v_{179} and v_{421} may project and interfere.

We have fleshed out the discussion of the eKE dependence of the beat frequencies in the main paper. The first paragraph of the discussion now reads:

... The beat frequency dependence upon eKE reveals that vibrational excitation of the S_1 state is preserved in v_{179} upon ionization and confirms that the 180 cm⁻¹ beat frequency is largely due to interference between multiple excited levels of v_{179} . Beats involving the $v = 1$ level of v_{421} , however, only appear at the highest eKE as the $v = 0$ level of the D_0^+ state is the common final state onto which all vibrational modes may project. The absence of beat frequencies larger than ~250 cm⁻¹ is due to the temporal resolution of our experiment (see Supplementary Figure 11 in the Supplementary Methods). Further discussion of the photoelectron structure and eKE dependence of the quantum beats may be found in Supplementary Discussion 1.

Furthermore, in our new discussion on quantum beat phase (see Author Response to comment 5), we have added the following text regarding the difference in eKE:

...The origin of this phase factor and the experimentally observed quantum beat phase may be understood by considering the appearance of the 180 cm⁻¹ beat at lower eKE. While the energy shift is related to the projection of v_{179} into the vibrationally excited cation (as discussed above), the diagonal beat recurrences shown in Supplementary Figure 7 show that quantum beats appearing at lower eKE also appear to be shifted in time delay. Such a temporal offset may be described as a quantum beat (namely the 180 cm⁻¹ beat) with a different phase.¹⁵ As this phase is tied to a change in eKE, the phase factors added to the time-autocorrelation function must serve to independently shift the time-autocorrelation “probe window” along each vibrational coordinate involved in a given beat. ...

3. In their time-domain analysis, the authors use a common damping time for all the vibrational frequencies. It seems counterintuitive to me that the dephasing time is independent of vibrational frequency. Have the authors considered using different dephasing times for different frequencies? Alternatively, can the authors provide a brief explanation for why dephasing times should be independent of vibrational mode?

Author Response and Action: The reviewer raises a critical point here and asks important questions regarding our analysis. A more detailed analysis of the vibrational frequencies may yield frequency dependent decay times; indeed the 330 nm FFT(Δt) spectrum shown in Figure 3 of the manuscript appears to show the 115 cm⁻¹ beat frequency disappearing on a shorter timescale than the other

frequencies. However, with our present data, it is not clear to what extent this behaviour is due to dephasing versus loss of the wavepacket via IVR. Additionally, while Equations S3 and S4 could be modified to include frequency-dependent dampening lifetimes, the resulting equations are over-parameterised and fits to the one-dimensional transients often do not converge. Furthermore, we do not wish to “over analyse” our data and make more detailed claims given the limited time windows we were able to observe. Thus, while it is a simplistic model, we have elected to only present an overall dampening lifetime for these data.

We have revised the discussion around fitting models in the Supplementary Methods, adding the following text after Equation S4:

As an alternative to Eqs. (S3) and (S4), the beat dampening time τ_{beats} could reasonably be beat-dependent (i.e. indexed by j). Indeed, as seen in Figure 3 in the main paper, the different beat frequencies appear to dampen with different lifetimes. However, based on the results at $\lambda_{pu} = 344$ nm and given the difficulty in differentiating between dephasing and IVR, the single “global” beat decay time was chosen.

4. In the Supplementary Information, it would be useful for the authors to list the frequencies extracted from their fits to eq. S3 (or S4), along with the amplitudes and phases.

Author Response and Action: We have added Supplementary Table 1 to the SI which includes the results of fits to one-dimensional MA transients at 348, 344, and 330 nm as well as MenA at 330 nm.

5. For the non-experts, the authors may wish to briefly describe the physical significance of the time-autocorrelation function. On a related note, Figure 3d shows the plot of the computed autocorrelation function. However, shouldn't it be the modulus-squared of the autocorrelation function – the survival probability – that should be compared to the experimental time-dependent photoelectron signal instead?

Author Response and Action: While both the modulus and the modulus-squared of the time-autocorrelation function appear to be used in the literature, the reviewer is correct that the modulus-squared of the time-autocorrelation function – and the interpretation as the wavepacket survival probability – is the correct form for the present analysis. Furthermore, we would like to thank the reviewer for drawing our attention back to the wavepacket calculations. Reviewing our work, we were able to more deeply explore the role of phase in these calculations, resulting in better agreement with experiment and a deeper understanding of the excited state photochemistry of MA.

We have updated Figure 4 as well as Supplementary Figures 11 and 12 to reflect the correct calculation. Additionally, for added transparency and reproducibility, we have included the specific data used to calculate the autocorrelation function as Supplementary Tables 2 and 3. We have also added text on page 7 of the manuscript regarding the interpretation of the time-autocorrelation function. The text now reads:

To aid the interpretation of these transients, the time-autocorrelation function of Eq. (1) was calculated¹⁵ using the dominant transitions observed in the LIF spectrum; see Supplementary Methods for further details. Essentially, the time-autocorrelation function describes the temporal behavior of the wavepacket as it moves out of and back into the Franck-Condon region. Plots of the survival probability of the time-autocorrelation function, i.e. $|\langle \Psi(\Delta t) | \Psi(0) \rangle|^2$, may serve as an initial comparison to

experimental transients,^{26,27} an example of the calculated “transient” is shown in Figure 4 d. ...

Additionally, based on our present wavepacket calculations, we have shifted our discussion to address the issues of phase and wavepacket projection in our experiment. We have added Supplementary Figures 7 and 8, expanded the description of the wavepacket calculations in the Supplementary Methods and added Supplementary Discussion 2. The last few paragraphs of the Discussion have also been significantly altered; the text now reads:

... Indeed, the quantum beats at this wavelength show clear evidence for dephasing and revival with the time-autocorrelation calculation in qualitatively good agreement with the revival times of each frequency. The most significant disagreement, an extra revival of the 60 cm^{-1} beat, is likely due to inaccuracies in the phase factor added to each vibrational level in the calculation. As discussed further in Supplementary Discussion 2, the inclusion of a phase factor into the wavepacket calculation is crucial to the reproduction of the experimental transients and observed quantum beat phase (see Supplementary Table 2).

The origin of this phase factor and the experimentally observed quantum beat phase may be understood by considering the appearance of the 180 cm^{-1} beat at lower eKE. While the energy shift is related to the projection of ν_{179} into the vibrationally excited cation (as discussed above), the diagonal beat recurrences shown in Supplementary Figure 7 show that quantum beats appearing at lower eKE also appear to be shifted in time delay. Such a temporal offset may be described as a quantum beat (namely the 180 cm^{-1} beat) with a different phase.¹⁵ As this phase is tied to a change in eKE, the phase factors added to the time-autocorrelation function must serve to independently shift the time-autocorrelation “probe window” along each vibrational coordinate involved in a given beat. Furthermore, the inclusion of a phase factor into these wavepacket calculations that depends on specific vibrational modes implies that a majority of the modes contributing to the 180 cm^{-1} beat are observed away from their respective classical turning points (i.e. the potential energy walls of each vibrational mode, see reference 35 for example) in the experiment. While the mixed nature of the S_1 vibrational states (as evidenced by the large number of satellite peaks near the main LIF progressions) makes firm assignment of each vibrational phase untenable at present, we may yet unveil some information about the S_1 state. A reasonable wavepacket reproduction of the experimental transient is obtained when the levels of ν_{179} are given a phase factor of $\pi/2$ relative to the other vibrational modes (as shown in Figure 4 d). This ν_{179} phase factor is different from applying a phase of $\pi/2$ to any $\sim 180\text{ cm}^{-1}$ beat, as several other combinations of LIF peaks also contribute to this beat (most significantly the LIF peak at 366 cm^{-1} and the cluster of peaks near 600 cm^{-1} shown in Supplementary Figure 2). Indeed, as shown in Supplementary Figure 8, wavepacket calculations with a phase factor of $\pi/2$ added to these modes result in substantially poorer agreement to the experimental 344 nm transient. It thus appears that the ionization window along the ν_{179} coordinate is shifted relative to the other normal modes.

Regardless of the nature of the observed quantum beat phase, the time-autocorrelation function is too simplistic to accurately model the MA beat behavior: this function does not contain any information about the S_1 potential energy landscape or an explicit projection onto the D_0^+ surface, factors which must play a significant role in the observed temporal behavior of these beats. A more accurate representation of photoexcitation and time-dependent photoionization of MA (such as through direct dynamics calculations) may reproduce the observed beat behavior^{36,37} and would be of significant interest. Thus, there is a clear need for additional theoretical and experimental work...

REVIEWERS' COMMENTS:

Reviewer #2 (Remarks to the Author):

The authors have thoroughly addressed the comments raised earlier by the Reviewers. I recommend publication of the revised manuscript in its current form.